# Physicochemical Properties of Membrane Adsorber from Palm Empty Fruit Bunch (PEFB) by Acid Activation

**DOI:** 10.3390/membranes11120917

**Published:** 2021-11-24

**Authors:** Nur Hidayah, Muthia Elma, Putri Vidiasari Darsono, Isna Syauqiah, Angelica Amenia, Daniel Guntur Laksana Putra, Heru Renaldi Akbar, Nurul Huda, Aulia Rahma

**Affiliations:** 1Department of Industrial Engineering, Faculty of Science and Engineering, Sari Mulia University, Jl. Pramuka No. 2, Banjarmasin 70238, Indonesia; 2Department of Chemical Engineering, Faculty of Engineering, Lambung Mangkurat University, Jl. A. Yani KM 36, Banjarbaru 70714, Indonesia; isnatk@ulm.ac.id; 3Materials and Membranes Research Group (M2ReG), Lambung Mangkurat University, Jl. A. Yani KM 36, Banjarbaru 70714, Indonesia; angelicamenia@mhs.ulm.ac.id (A.A.); DanielGLP@mhs.ulm.ac.id (D.G.L.P.); HeruRA@mhs.ulm.ac.id (H.R.A.); Hudanurul@mhs.ulm.ac.id (N.H.); arahma@mhs.ulm.ac.id (A.R.); 4Pharmacy Department, Faculty of Health, Sari Mulia University, Jl. Pramuka No. 2, Banjarmasin 70238, Indonesia; putrividiasari@gmail.com

**Keywords:** palm empty fruit bunch, membrane adsorber, chemical activation, HCl

## Abstract

A membrane adsorbent was successfully made from palm empty fruit bunches (PEFB), which was pyrolysed as physical activation. The effect of adding the impact of one-step catalyst (hydrochloric acid) and differences in the concentration on the characteristics and structure and deconvolution are investigated in this study. The results of the research have been successfully created and characterised using Fourier-Transform Infrared (FTIR), X-ray diffraction (XRD), Brunauer–Emmett–Teller (BET) isotherm, and membrane morphology using SEM test. Membrane performance testing was carried out using a biogas flame test. The adsorber membrane was made by adding NH_4_Cl as a cationic surfactant, polyvinyl acetate (PVA), and polyethylene glycol (PEG) with a ratio of 1:3. The FTIR test has a functional group: O-H; C-H stretch; C=C-C; Arly O-Strech; C-O. Adsorbent membrane with the addition of 0.5 M HCl catalyst had the highest ratio of O-H/C=C-C relative area of 4.33. The diffractogram shows an amorphous structure with (002) and (100) graph planes. Adsorber membrane with a concentration of 1.5 M HCl has formed amorphous structured fibre. The adsorber membrane with a concentration of 0.5 HCl activator gave a surface area of 0.5345 m^2^ g^−1^ and a pore volume of 0.000983 cm^3^ g^−1^.

## 1. Introduction

In recent decades, clean and renewable energy has become an interesting topic among researchers. Biogas is one of the products resulting from the bioconversion of organic materials used as a renewable energy fuel [1]. Biogas is generally produced from methanation of biomass and organic waste from anaerobic digestion sewage sludge, commercial compost, landfill, biomass gasification (thermochemical production process), anaerobic livestock manure with energy systems, and agro-food industrial processing facilities in mesophilic conditions (35 °C) and thermophilic conditions (55 °C). The composition of biogas generally consists of 45–55% CH_4_, 30–40% CO_2_, 5–15% N_2_, and 0–3% H_2_, 0–3% CO, 0–5% O_2_, 1–5% H_2_O, 20–200 ppm halogenated hydrocarbons, 0–100 ppm H_2_S and 0–5 ppm NH_3_ [2]. Of course, various kinds of contaminants in landfill biogas have a good and bad influence on the quality of the biogas produced. Abatzoglou and Boivin (2009), in their journal, said that components such as NH_3_, sulfuric acid (H_2_SO_4_), and siloxanes are impurity contaminants that can cause corrosive, health risks, environmental damage, and are dangerous during the combustion process because they can form microcrystalline silica [3].

However, not all contaminants in biogas are harmful; other components such as CO_2_, H_2_O, O_2_, and N_2_ are considered valuable, as well as O_2_, which can help in some H_2_SO_4_ removal technologies. However, suppose the contaminant level is higher (almost equal to about 95% of the methane content). In that case, it can reduce the energy content of the biogas and result in a less than optimal quality of biogas production due to its low heating value [2]. Biogas with high impurity gas components such as CO_2_, H_2_O, and H_2_S can cause a decrease in calorific value. Therefore, biogas purification efforts are used to remove contaminants and achieve a distinctive biogas composition so that a high calorific value is achieved on combustion.

The biogas purification process can be carried out in several ways or methods, depending on the gas flow rate and economy [4]. According to Ryckebosch et al. (2011), biogas purification can be carried out in two main stages, namely removing trace components such as hydrogen sulphide (H_2_S) and water vapour (H_2_O) which can cause corrosion in equipment and removing carbon dioxide gas to increase the heating value [5]. In general, the biogas purification process can be carried out physically, chemically, and biologically. Physical purification can be done by the adsorption method [6,7]. Biological purification uses bacteria as a decomposer of impurities. In comparison, chemical purification is one of them by using the activated absorption method [6]. Several purification technologies have been developed, including water scrubbing, organic solvent scrubbing, pressure swing adsorption, cryogenic separation process, chemical hydrogenation process, in situ and ex situ biological upgrading, chemoautotrophic methods, membrane separation, and others [1,2,5,8,9,10,11,12]. Membrane technology has several advantages: the separation process takes place at room temperature, can be carried out continuously, is variable, can be adjusted as needed, and the resulting membrane can be reused and is environmentally friendly because it does not cause a destructive impact on the environment. The membrane can also function as a specific filter because only certain molecules can pass through the membrane while the rest will be stuck on the surface [13].

There have been many studies regarding the purification of biogas by the membrane method [8,14,15,16,17,18]. Although the application is relatively easy, separating the membrane method weakens the wet or dry washing process. To overcome this, research has been carried out to integrate the membrane and adsorption based on the working principle of the filtration membrane and the adsorbent column. The union of these two methods is called the membrane adsorber [19]. The adsorber membrane is helpful in adsorption applications (purification) because it has a good retention capacity, especially at high loading rates. In addition, adsorber membranes require low production costs. In its operation, the adsorber membrane can capture molecules with a size of <200 kDa [20]. In the adsorber membrane, the type of adsorbent to be used as a raw material for the membrane has an important influence on capturing impurity gas or purified gas. These different types of adsorbents will provide other functional groups and adsorption properties of the membrane [19].

Indonesia is the world’s number one producer of palm oil, with a total production of 48.4 million tons in 2019. This amount of production increased by 12.92% compared to the previous year, with sixty million hectares [21]. The value of Indonesian palm oil exports in 2019 reached USD 15.55 billion. Palm oil exports are the largest foreign exchange earner for the export sector of industrial products [22]. A large amount of waste also accompanies the promised economic value of the palm oil industry from the process. Processing one ton of fresh fruit bunches produces palm oil mill effluent (POME) as much as 583 kg [23].

Meanwhile, the types of waste at t generated include mesocarp fibre (MF), shells, and empty bunches of 144 kg, 64 kg, and 210 kg, respectively. PEFB are the most considerable solid waste produced in this process [24]. PEFB has a high biomass content. However, the utilisation of PEFB is limited to a mixture of animal feed, fertiliser, boiler fuel, or even just buried. Improper handling of PEFB can cause air pollution from leachate and carbon dioxide gas emissions from the combustion process [25]. Therefore, efforts are needed to utilise this waste, and its abundance becomes a more promising product.

Palm empty fruit bunch contains many nutrients, organic matter and physically consists of various kinds of fibre with a composition of about 36.81% cellulose, 27.01% hemicellulose, and 22.60% lignin [26]. With high organic content, palm empty fruit bunch can be used as activated charcoal in the separation process using the adsorption method. There have been many studies that have proven the effectiveness of adsorbents from oil palm empty fruit bunches [27,28,29,30,31,32,33,34,35].

The test results of activated carbon of oil palm empty fruit bunches show the presence of carbon groups and surface area with an average BET test of more than 500 m^2^/g [27]. This supports the use of PEFB-activated carbon as raw material for membrane manufacture. It was reported that PEFB waste had been used as a membrane matrix to absorb motor exhaust gases. PEFB waste is first synthesised into activated carbon and cellulose acetate and then mixed to form a membrane matrix [36]. Another study also reported PEFB activated carbon as an additive for the manufacture of ceramic membranes. It is suspected that the addition of activated carbon from oil palm empty fruit bunches can increase the surface area in the BET test [37].

This study aimed to prepare and characterise membrane adsorber from empty palm fruit bunches using a one-step acid-base catalyst. This study used physical-chemical activation methods such as physics using pyrolysis. In contrast, chemical activation was produced through chemical activators using hydrochloric acid (HCl) concentrations of 0.5, 1, and 1.5 M. The effectiveness of membrane adsorber will be seen from the Fourier-Transform Infrared (FTIR), X-ray diffraction (XRD), Brunauer–Emmett–Teller (BET) isotherm, and membrane morphology using the SEM test.

This work shows that adding an acid catalyst to the adsorber membrane for activated carbon extract of palm empty fruit bunches can improve the performance of the absorption effectiveness through the formation of a larger absorption area compared to a membrane that only undergoes physical activation. This work determines the possible application of adsorber membranes in the gas separation process, especially biogas. The performance of the membrane was seen through the FTIR test, N_2_ physisorption, to determine the surface area and pore diameter. XRD and SEM analyses were conducted to see the formation of amorphous structures.

## 2. Materials and Methods

### 2.1. Chemical and Materials

OPEFB membrane adsorber was prepared using chemicals and materials such as palm empty fruit branches (PEFB) from PT. Kharisma Alam Persada, South Kalimantan, Indonesia; isopropyl alcohol (2-propanol, Merck, Singapore); ammonium chloride (NH_4_Cl, Merck); polyethylene glycol 400 (PEG 400); polyvinyl alcohol (PVA); filter paper; and deionised water (DI water Merck).

### 2.2. Preparation and Activation of Activated Carbon PEFB

Palm empty fruit bunches adsorber membrane was made by pyrolysis process as physical activation and hydrochloric acid (HCl) as a chemical activator. PEFB were collected from oil palm plantations that have been washed by freshwater and dried for seven days under the sunlight. After a week, the PEFB were physically activated using pyrolysis apparatus at 500 °C for 30 min. Subsequently, the activated PEFB was mashed and sieved with 200–400 mesh to obtain the uniform extent. This powder can be called a physical activation adsorber. The activation process was done by adding hot distilled water to 3 beaker glasses which already contained 20 g of PEFB powder until all parts were wet.

Furthermore, the activator solution used is hydrochloric acid (HCl), added 100 mL with variations of 0.5, 1, and 1.5 M in each beaker and evaporated to dry on a hot plate at a temperature of 120 °C. After that, we pour 200 mL of distilled water into the three containers and leave it on a hot plate for 30 min at 50 °C. Next, it is filtered with Whatman filter paper and washed 15 to 20 times using distilled water until normal. The result of filtering is in the form of a solid residue. Then, the powder is dried in an oven at 60 °C, and the activated powder is weighed. Schematic set-up fabrication of membrane adsorber PEFB can be seen in Figure 1, as follows.

### 2.3. Preparation and Characterisation of Membrane Adsorber PEFB

The powder was made with hydrochloric acid variations of 0.5, 1, and 1.5 M. It was put into each 500 mL beaker, then mixed with 35 mL of isopropyl alcohol stirred using a magnetic stirrer at a speed of 600 rpm for 10 min, after which it is filtered. Then, the filter results were added to 3.5 g of NH4CL as a cationic surfactant, dissolved in 300 mL of distilled water. The mixture was then stirred with a magnetic stirrer at a speed of 600 rpm for 1 h to form nano-sized membrane pores (Chowdhury et al., 2006) and filtered using a vacuum pump to leave a solid residue. Solids with variations of 0.5, 1, and 1.5 M in the container were then added respectively to 3.4 g of PVA and 5 mL of PEG, and the remaining 8 mL of filter solution were added gradually until the membrane mixture was solid and fully adhered. The membrane mixture was printed on a membrane mould made of stainless steel with a diameter of 7 cm and a height of 0.6 cm. Then, the membrane was dried in the sun for about one day. After the upper surface of the membrane was slightly dry, the membrane was then pressed using a hydraulic jack with a pressure of 100 psi.

FTIR (Fourier-Transform Infrared) was used to investigate the chemical properties of silica-carbon xerogels. FTIR spectra data were collected from FTIR type Bruker Alpha. Instrument type: alpha sample compartment RT-DLaTGS accessory: ATR platinum Diamond 1 Relf. The spectra were collected from 30 scans ranging between wavelengths of 500–4000 cm^−1^. Peak deconvolution of the absorption bands over the region 1300–700 cm^−1^ was performed with Fityk software using Gaussian line shapes with a least-square fit routine. Peak areas were measured for the normalised spectra using a local baseline [38]. X-ray diffractometer (XRD) with the Copper anode (Cu-Kα, λ = 1.5406) was used to identify the formed phases in the membrane adsorber. The morphology of membrane adsorber can be detected through the SEM-EDS method. The sample is placed on the surface of the holder who has been given carbon tape, and then gold is coated on the surface of the model. After coating, the sample is inserted in the sample room, which has been vacuumed first. The selection is analysed using an SEM instrument (HITACHI FLEXSEM 100) to find out the morphology. Nitrogen physisorption analyses were conducted using a Micromeritics TriStar 3000 instrument. The sample was degassed under vacuum for six h at 200 °C. The specific surface area was determined from the Brunauer, Emmett, and Teller (BET) method. The Dubinin–Astakhov and Barrett–Joyner–Halenda methods were taken to determine the average pore sizes of microporous and mesoporous materials, respectively.

### 2.4. Biogas Purification Using Membrane Adsorber

Biogas is flowed from the modified plastic of the temporary biogas reservoir (1) to the compressor to be accommodated. The next biogas from the compressor (2) is pumped to the rotameter (3). The rotameter flow rate is 0.025 L/s (Iriani and Heryadi), the rotameter valve is opened to flow biogas into the adsorber column (4) until the biogas passes through the filtration membrane of the adsorber. Furthermore, the biogas that has passed through the filtration is directly accommodated in a 0.25-L Tedlar bag (5) and labelled, which is carried out three times with membranes that have been varied. The gas that has been stored in a Tedlar bag is ready to be tested visually with three repetitions of the flame test. The set-up can be seen from Figure 2:

## 3. Results and Discussions

The FTIR spectra of unmodified membrane adsorber without HCL activation (AC) and activated membrane adsorber using g 0.5 (H1), 1 (H2), 1.5 (H3) M HCl are presented in Figure 3. It can be seen in Figure 3 that the spectrum of the activated carbon adsorber membrane without the addition of an HCl activator has a flatter peak than the activated carbon adsorber membrane after the addition of HCl. The elevation found in the ~3382 cm^−1^ wave represents the O-H stretching vibration in the hemicellulose group. Compared to AC, the O-H peaks for H1, H2, and H3 are wider, increasing the O-H stretching functional group and water absorption on the surface. This result is in line with a similar study by Tan et al., 2017, which has examined coconut-shell-based activated carbon, where O-H stretching implies the presence of more O-H groups from an increase in carboxylic groups on activated carbon that has been modified by HCl acid [39].

Meanwhile, the wave ~2869 cm^−1^ corresponds to the C-H stretching mode as a function of the cellulose group. The same trend also shows H1, H2, and H3 peaks are more prominent. This shows the sample reacts differently in different concentrations of HCl. The functional group C=C-C, where the entire waveform is defined as the aromatic ring strain, was observed at ~1607 cm^−1^. The wavelength at ~1237 cm^−1^ belongs to the O-aromatic ether strain. The sharp hydroxyl band occurs around ~1077 cm^−1^. The functional groups in unripe palm empty fruit bunches (PEFB) have a complex and straightforward spectrum. This has been studied further by A. R. Hidayu et al. (2013), which showed an adsorption peak at 3302 cm^−1^ and identified as O-H stretch, which in this case indicated the presence of a bound hydroxide compound in crude palm empty fruit bunches (PEFB) [28]. The distribution of wave crests is also found at a wavelength of 1739 cm^−1^ which is indicated as a C=O functional group, followed by the wavelength in the area of 1216 cm^−1^ and 1032 cm^−1^, which also shows the C-O stretch active group (Lu et al., 2007) [40].

The results of the Fityk graph in Figure 4 show differences in adsorber membrane from activated carbon of PEFB before and after the addition of the activator. It can be seen that the highest peak area ratio was obtained at the chemical activation of 0.5 Molar HCl for the O-H functional group. The O-H functional group represents the hydroxide [41]. At the same time, the lowest peak area ratio is at 0.0 Molar of physical activation. In comparison, the lowest peak area ratio is on the adsorber membrane with a concentration of 0.0 Molar HCl. The hydroxyl group rises on the carbon because the carboxylic group is added. The HCl treatment increases the oxygen in the carbon content as well. This leads to an increased adsorption capacity of activated carbon. Instead of forming new functional groups, high concentrations of HCl convert carbonyl or carboxyl groups into phenol and lactone groups [42]. Previous studies by Hidayu et al., 2013, reported that the specific surface area of activated carbon will increase with increasing acid concentration [28].

Figure 4 shows adsorber membrane from activated carbon of PEFB without chemical activation has the most little OH stretching strain. In contrast, the adsorber membrane that chemical acids have activated shows that the increase in OH is caused by the rise in the acid impregnation ratio; the acid contained in PEFB causes a higher carbon combustion rate [29]. This is in line with previous studies, on the outer surface of the O-H increasing rapidly, which is a domain factor that affects adsorption efficiency. This occurs because the strain on physical activation is not visible or not formed, while through physicochemical activation, functional groups are included, and the peaks are formed are broader [34,43]. However, the ratio of at 1 M HCl concentration was lower than without activation and the other two HCl concentrations.

Based on Table 1, the adsorber membrane from activated carbon of PEFB without chemical activation has a small O-H area, compared to the adsorber membrane from activated carbon of PEFB with chemical activation of HCl (0.5; 1; 1.5 M). This is because the presence of the O-H functional group is caused by the addition of the acid compound—in this study, HCl [29]. Adsorber membrane from activated carbon of PEFB with variations in the acquisition of chemical concentrations of HCl has different peak area values for O-H. The lowest OH peak area value was found in the 1 M HCl chemical activation. This was inversely proportional to the C=C-C peak area (aromatic strain ring), where the 1 M HCl chemical activation sample showed the highest value compared to the activation samples 0, 0.5, and 1.5 M HCl. When viewed from the value of the ratio of O-H/C=C-C, the value of the 1 M HCl chemical activated PEFB adsorber membrane sample has the lowest number of a 2.13 unit area.

The results of the XRD graph shown in Figure 5a above show that the adsorber membrane sample made has an amorphous shape. This is shown from the pattern results on the XRD of the OPEFB adsorber membrane without activation, which obtained an irregular shape, which is an amorphous characteristic. Although no specific peaks were produced, 2θ angles that could be read were created, namely 14–16°, 20–24° and 25–27° which were 2θ angles of legible activated carbon. Based on Figure 5b, it can be seen that the X-ray diffraction pattern or the increase in the crystallinity value between variations with one another has an insignificant difference, indicating the presence of amorphous carbon that is irregularly stacked by the carbon ring [27]. Thus, the activated carbon contained in the membrane can be identified as carbon with an amorphous structure. There are massive microcrystals in which two distinct diffraction peaks represent the respective diffraction peaks characteristic of the microcrystalline crystalline surface [44].

The results of SEM characterisation were used to observe the morphology of the adsorber membrane that had been physically activated and physicochemically activated with variations in HCl content: 0.5, 1, and 1.5 Molar at a magnification of 5000 times shown in Figure 6. Figure 6a,b show that uneven carbon and polymer content causes polymer clumps so that the membrane does not mix well. The addition of concentration materials gave a significant difference to the mixture of membrane-making materials. From the results of the SEM characteristics, it can be seen that the carbon pores formed on the surface are very irregular, and not too many pores are formed. Based on the XRD test, the adsorber membrane has an amorphous structure. The SEM test on the PEFB adsorber membrane activated a physics-chemical concentration of 1.5 M HCl, forming amorphous structured fibre.

In the International Union of Pure and Applied Chemistry (IUPAC) classification, pores are micropores (50 nm diameter). This classification is essential because most molecules of gaseous pollutants vary from 0.4 nm to 0.9 nm in diameter. Gas-phase activated carbons usually consist predominantly of micropores, whilst liquid-phase activated carbons have significant mesopores because of the larger sizes of liquid molecules. Figure 7 shows the adsorption-desorption process of nitrogen from adsorbing activated carbon OPEFB, which belongs to the type 1 isotherm. Type 1 isotherm causes a flat or almost flat-convex curve where the adsorption isotherm directly intersects the line P/Po = 1.

The surface area of all test variables on the PEFB adsorber membrane is given in Table 2. The highest surface area calculated using the BET method was found in the variable with a 0.5 M HCl activator concentration of 0.5345 m^2^/g. Inversely proportional to the pore volume and the average pore diameter produced, the variable with 0.0 M HCl activator concentration has the most significant values, namely 0.00878 cm^3^ g^−1^ and 130.13 nm. In this study, the OPEFB adsorber membrane with an activating agent concentration of 0.5 M had the lowest pore volume and average pore diameter compared to other variables due to the presence of acidic substances that inhibited the formation of pores, especially micropores. The surface area and pore volume values obtained can be said to be relatively small. This can be caused by a polymer material, which only functions as an adhesive and causes clumping to inhibit pores’ formation.

Based on Figure 8, visible changes in the colour of the flame indicate an increase in performance in biogas which has been purified using a membrane. Qualitatively, the pure biogas flame test results are predominantly yellowish red, which indicates that pure biogas has less CH_4_ content than CO_2_ and other impurities, which have a higher value. The colour changes on activation of the physicochemical variation of HCl in Figure 8 look very similar in terms of the colour produced. In the physicochemical variation with a content of 0.5, 1, and 1.5 M, the dominant flame colour is bright blue compared to physical activation. The chemical activation of 1 M HCl looks fire-coloured with a clear blue spectrum compared to 0.5 and 1.5 M and looks stable. Flames with a dark blue spectrum have a high calorific value, and the burning of biogas with blue flames is the result of methane gas. The biogas flame test showed that the oil palm empty fruit bunch adsorber membrane has potential for application in gas purification, although the BET test results show the formation of a pore size that is not too large.

## 4. Conclusions

A palm empty fruit bunch adsorber membrane has been successfully fabricated through chemical activation and characterised using FTIR test, XRD test, and membrane morphology using SEM test and nitrogen absorption test using BET method. In the FTIR test, there are functional groups, namely O-H; C-H stretch; C=C-C; Arly O-Strech; C-O, which indicates that activated carbon has been formed. The most significant number of activated carbon group formations occurred in the addition of a 0.5 M HCl activator. Based on the XRD test, the adsorber membrane has an amorphous structure. Based on the SEM test on PEFB, the adsorber membrane activated by physicochemical concentration of 1.5 M HCl has a fibre shape and asymmetric design. XRD and SEM tests showed that the adsorber membrane has a slightly amorphous structure that forms fibres and an asymmetrical structure with a dominant crystal shape. Judging from the surface area and pore volume in the nitrogen isotherm absorption test, all the variables tested showed small values. This is probably caused by the addition of PAV and PEG polymer materials as adhesives which are still not appropriate. The surface area of the nitrogen isotherm is found in the variable with a concentration of 0.5 M HCl, which is 0.5345 m^2^/g. As for the size of the pore volume and the average pore diameter, the most significant results were obtained in the variable with an HCl concentration of 0.0 M. The most significant surface area test values were 0.5 M, respectively; 1 M; 1.5 M; and 0.0 M. For the results of the pot volume and average pore diameter, respectively, the most significant yield was 0.0 M; 1 M; 1.5 M; and 0.5 M.

## Figures and Tables

**Figure 1 membranes-11-00917-f001:**
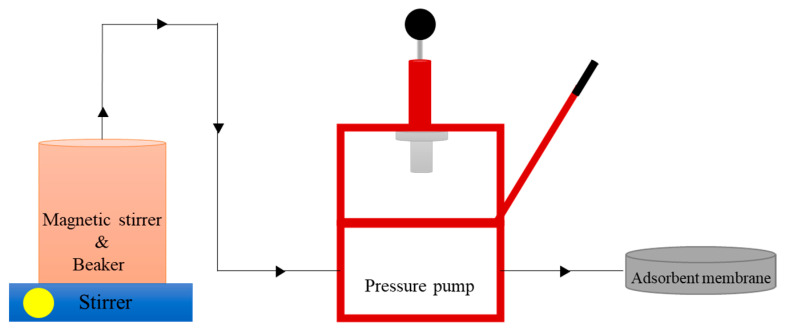
Schematic set-up fabrication of membrane adsorber PEFB.

**Figure 2 membranes-11-00917-f002:**
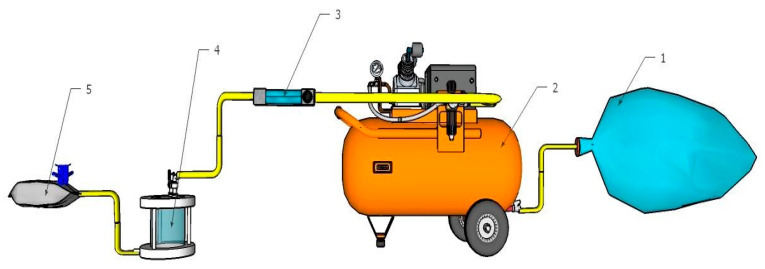
Set-up of biogas purification process.

**Figure 3 membranes-11-00917-f003:**
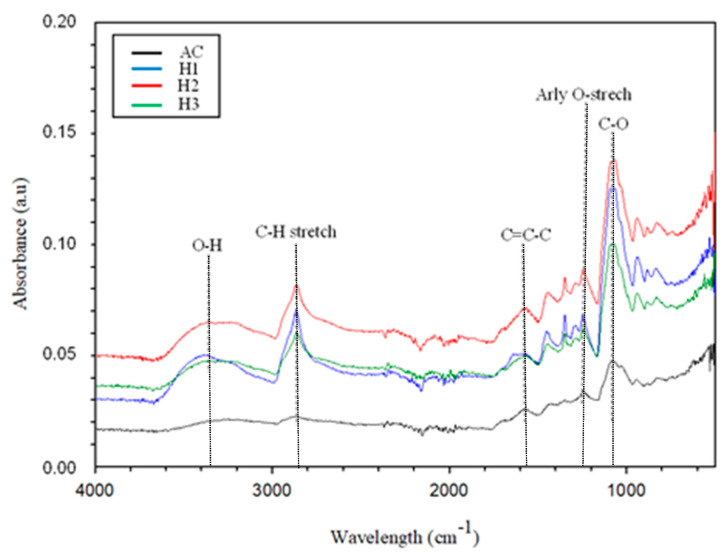
FTIR spectra of adsorber membrane with Activated Carbon (AC), (H1) HCl 0,5 M; (H2) HCl 1 M; (H3) HCl 1.5 M.

**Figure 4 membranes-11-00917-f004:**
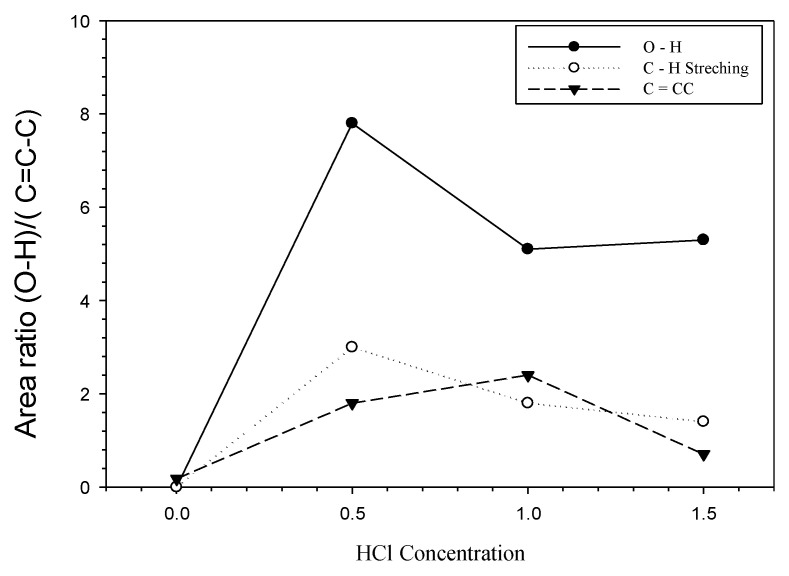
Deconvoluted peak area ratios of activated carbon concentration at adsorber membrane.

**Figure 5 membranes-11-00917-f005:**
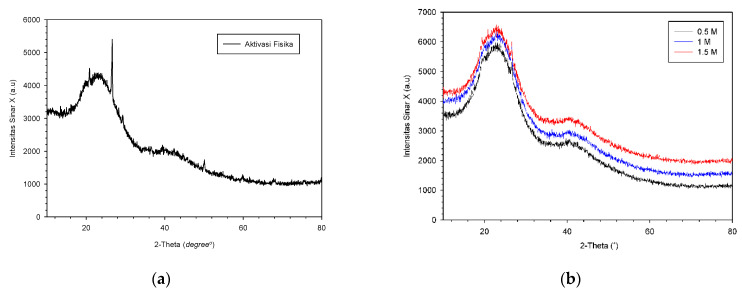
Diffractogram XRD of (**a**) unmodified membrane adsorber without HCL activation (AC); (**b**) activated adsorber membrane using g 0.5 (H1), 1 (H2), 1.5 (H3) M HCl.

**Figure 6 membranes-11-00917-f006:**
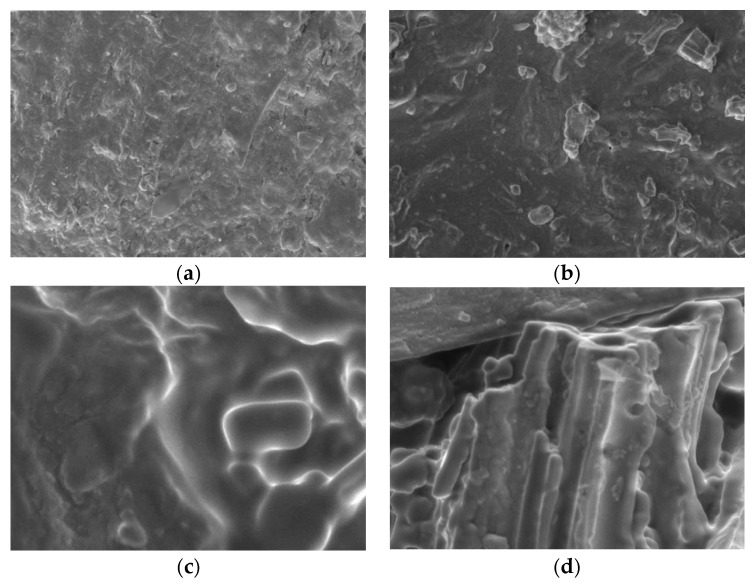
SEM morphology of: (**a**) unmodified membrane adsorber without HCL activation (AC); (**b**) activated membrane adsorber using g 0.5 M; (**c**) 1 M; (**d**) 1.5 M HCl.

**Figure 7 membranes-11-00917-f007:**
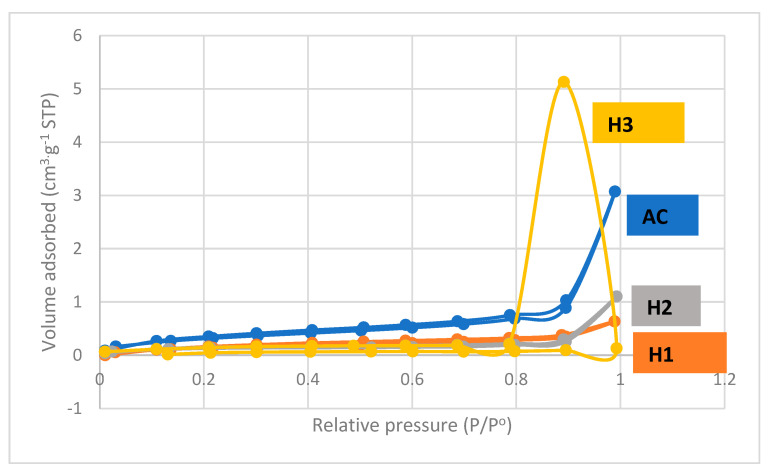
Plots of biogas isotherm data of unmodified membrane adsorber without HCL activation (AC) and activated membrane adsorber using g 0.5 (H1), 1 (H2), 1.5 (H3) M HCl.

**Figure 8 membranes-11-00917-f008:**
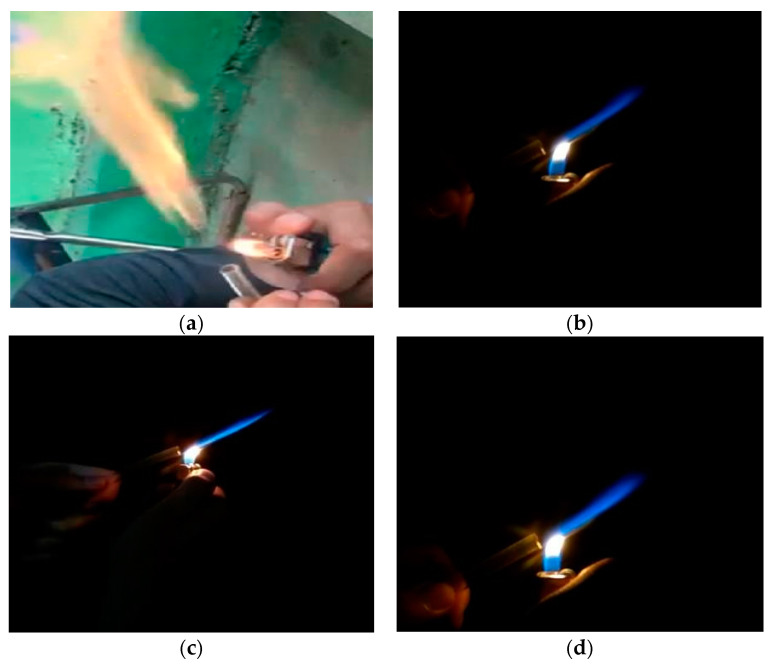
Flame test (**a**) without using a membrane, (**b**) 0.5 M HCl, (**c**) 1 M HCl, and (**d**) 1.5 M HCl PEFB adsorber membrane.

**Table 1 membranes-11-00917-t001:** Deconvolution of O-H/C=C-C concentration.

HCl Concentration	Area (Qn)		Area Ratio
	O-H	C=C-C	H/C=C-C
0	0.01	0.18	0.00
0.5	7.8	1.8	4.33
1	5.1	2.4	2.13
1.5	5.3	0.7	7.57

**Table 2 membranes-11-00917-t002:** Surface properties of palm empty fruit bunch membrane adsorber.

Concentration (M)	SBET (m^2^ g^−1^)	Pore Volume (cm^3^ g^−1^)	Average Pore Diameter (nm)
0	0.2699	0.00878	130.12732
0.5	0.5345	0.000983	7.34468
1	0.4387	0.001705	15.54718
1.5	0.4001	0.001484	14.83716

## Data Availability

Not applicable.

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
