# Peer review of "Physicochemical Properties of Membrane Adsorber from Palm Empty Fruit Bunch (PEFB) by Acid Activation"

_membranes, 2021, doi:10.3390/membranes11120917_

Round 1

Reviewer 1 Report

This manuscript developed activated carbon using Palm Empty Fruit Bunch (PEFB) as precursor and the prepared activated carbon PEFB was characterized using FTIR, XRD, BET and SEM. The authors also described the preparation of membrane adsorber in the experimental section on page 5 and they proposed to use the membrane adsorber for biogas purification process, but I did not see any results for this application. It seems some content is loss on page 9 or this manuscript is not completed. Overall, this manuscript is more about materials rather than membrane since there is no any reported membrane performance. From this point, this paper is not suitable for publishing on Membranes. In addition, there are a lot of typos in this manuscript (e.g. “Membran” in the title, “vacuum pupm” on page 5 line 7, “Table 2” should be Table 1 on page 7, “Figure 4” on page 8 should be Figure 5). These typos make the manuscript poor quality. Thus, I would not recommend publication, at least in its present form.

Author Response

Dear reviewer

Thank you for your advice on my article. It means a lot to me to continue to improve the quality of my articles. I apologize in advance if this article is late for you. I beg you not to reject my article. if there is something I need to fix again I will try to be better and will fix it as soon as possible

warm regards

Nur Hidayah

Reviewer 2 Report

The author studied physicochemical properties of membrane adsorber from PEFB. The topic is interesting.

However, there are too many typos. For example, in the title “membran”. This typo occurred in several locations in the paper. If you look at Fig.4, the low left should c) not b).

Above the Table 3, “Figure 5 show” , it seems that a paragraph is missing.

And page 2 is totally empty.

Suggest that the authors correct these errors before the reviewing.

Author Response

(The authors gave the same response as above.)

Reviewer 3 Report

This problem is relevant for journal scope. The manuscript follows the formal regulations of MDPI journals.

I suggest the major revision of the manuscript.

Remarks, suggestions, questions

  1. The main remark: please emphasize the novelty side(s) of your research.
  2. Please cite more papers from this journal at the last two years in the similar topic of this research.
  3. Please avoid the cumulated references, as it can be found in the Introduction part.
  4. Please introduce the parameters of XRF method.
  5. The Conclusion part is too short and general.
  6. What do you think about the chance for industrial application of your process?
  7. What is the unit of concentration in Figure 3?

Author Response

(The authors gave the same response as above.)

Round 2

Reviewer 1 Report

  1. The revised manuscript is shown in a change mode, which causes the manuscript hard to read. It would be better if the authors could list the revision and responses to the revewers' comments in a separated document. 
  2. Although the authors characterized the membrane adsorber using FTIR, XRD, BET and SEM, how do the physical/chemial proprties of membrane adsorber affect the adsorption performance? More discussion about the releationship between physical/chemial proprties and adsorption performance should be inculded in the revised manuscript.
  3. According to the BET results, all the prepared membrane adsorber showed very low surface area and small pore volume. Does this mean the preapred membrane adsorbers can not be used for practical application? The adsorption performance of the prepared membrane adsorbers should be measured. 

Reviewer 2 Report

apparently, the author used edit-tracking function. such a manuscript is not friendly for reviewers. the reviewer will not provide edit service. can you please clean up everything about edits in the manuscript and resubmit it?

Reviewer 3 Report

Thank you very much for your answers. I suggest the acceptance in this present form for publication.

Author Response

Dear reviewer
Thank you for your kindness in reviewing and providing suggestions so that our articles can be accepted for publication in the journal membranes

Round 3

Reviewer 1 Report

Authors replied to my comments accordingly. The manuscript now is recommended to be accepted.

Author Response

(The authors gave the same response as above.)

Reviewer 2 Report

can you please clean up your track edits and provide a complete manuscript without those edit track?

Author Response

Dear reviewer

First of all, let us thank you for being willing to review our manuscript.

Point 1: Can you please clean up your track edits and provide a complete manuscript without those edit track?

Response 1: As we said before, the use of the edit track function is a requirement of the journal editor team so we cannot remove this function in the manuscripts that we upload in the membranes journal. However, we have sent the manuscript without the edit tracking function in the cover letter to fulfil the reviewer's request. Reviewers can see on the next page in the cover letter we sent.

Round 4

Reviewer 2 Report

it would be better if the author can perform composition test and quantitative of biogas before and after purification with the membrane.

Author Response

Response to Reviewer 2 Comments

Dear reviewer

First of all, let us thank you for being willing to review our manuscript.

Point 1: it would be better if the author can perform composition test and quantitative of biogas before and after purification with the membrane.

Response 1: Yes, that's right, we agree with what the reviewer said. However, for now, we have only carried out qualitative tests to see the results of membrane performance. For compositional and quantitative testing using gas chromatography, it takes a longer time due to the pandemic conditions resulting in a large number of sample queues for the test, so it is not yet possible for us to carry out the test. I hope the reviewer can understand this and still allow this article to be published.